# Short term effect of antimicrobial photodynamic therapy and probiotic *L. salivarius* WB21 on halitosis: A controlled and randomized clinical trial

**Pamella de Barros Motta[1], Marcela Leticia Leal Gonçalves[1,2], Juliana Maria Altavista Sagretti Gallo[3], Ana Paula Taboada Sobral[1,2], Lara Jansiski Motta[1], Marcia Pinto Alves Mayer[4], Dione Kawamoto[4], David Casimiro de Andrade[5], Elaine Marcílio Santos[2], Kristianne Porta Santos Fernandes[1], Raquel Agnelli Mesquita-Ferrari[1], Alessandro Melo Deana[1], Anna Carolina Ratto Tempestini Horliana[1], Sandra Kalil Bussadori[1,6]***

**1** Post Graduation Program in Biophotonics Applied to Health Sciences, Universidade Nove de Julho (UNINOVE), São Paulo, SP, Brazil, **2** Post Graduation Program in Health and Environment, Universidade Metropolitana de Santos (UNIMES), Santos, SP, Brazil, **3** Post Graduation Program in Veterinary Medicine in The Coastal Environment, Universidade Metropolitana de Santos (UNIMES), Santos, SP, Brazil, **4** Department of Microbiology, Institute of Biomedical Sciences, Universidade de São Universidade Paulo, São Paulo, SP, Brazil, **5** School of Dentistry, University of Porto, Porto, Portugal, **6** Dentistry College, Universidade Metropolitana de Santos (UNIMES), Santos, SP, Brazil

\* sandra.skb@gmail.com

**Data Availability Statement:** All data files are available from https://www.ebi.ac.uk/biostudies/

## Abstract

### Objective

This study aimed to evaluate the effect of antimicrobial photodynamic therapy (aPDT) and the use of probiotics on the treatment of halitosis.

### Methods

Fifty-two participants, aged from 18 to 25 years, exhaling sulfhydride ($H_2S$) $\geq$ 112 ppb were selected. They were allocated into 4 groups (n = 13): Group 1: tongue scraper; Group 2: treated once with aPDT; Group 3: probiotic capsule containing *Lactobacillus salivarius* WB21 (6.7 x $10^8$ CFU) and xylitol (280mg), 3 times a day after meals, for 14 days; Group 4: treated once with aPDT and with the probiotic capsule for 14 days. Halimetry with gas chromatography (clinical evaluation) and microbiological samples were collected from the dorsum of the tongue before and after aPDT, as well as after 7, 14, and 30 days. The clinical data failed to follow a normal distribution; therefore, comparisons were made using the Kruskal-Wallis test (independent measures) and Friedman ANOVA (dependent measures) followed by appropriate posthoc tests, when necessary. For the microbiological data, seeing as the data failed to follow a normal distribution, the Kruskal-Wallis rank sum test was performed with Dunn's post-test. The significance level was α = 0.05.

### Results

Clinical results (halimetry) showed an immediate significant reduction in halitosis with aPDT (p = 0.0008) and/or tongue scraper (p = 0.0006). Probiotics showed no difference in relation

studies/S-BSST1378 and the DOI for the dataset is
10.6019/S-BSST1378.

**Funding:** This study was financially supported by
Fundação de Amparo à Pesquisa do Estado de São
Paulo (FAPESP) in the form of grants (2019/
14229-6 and 2022/16828-7) received by SKB. This
study was also financially supported by Conselho
Nacional de Desenvolvimento Científico e
Tecnológico (CNPq) in the form of grants received
by SKB (306577/2020-8), KPSF (304142/2023-9)
and RAMF (310491/2021-5). This study was also
financially supported by Coordination for the
Improvement of Higher Education Personnel
(Capes) (Postgraduate Program in Biophotonics
Applied to Health Sciences/Biophotonics Medicine)
in the form of a grant (PROEX 1294/2023) award
received by KPSF. This study was also financially
supported by the Coordenação de
Aperfeiçoamento de Pessoal de Nível Superior –
Brasil (CAPES) in the form of an award
(88887.818185/2023-00) received by PBM. The
funders had no role in study design, data collection
and analysis, decision to publish, or preparation of
the manuscript.

**Competing interests:** The authors have declared
that no competing interests exist.

to the initial levels (p = 0.7530). No significant differences were found in the control appointments. The amount of *Porphyromonas gingivalis*, *Tannerella forsythia*, and *Treponema denticola* were not altered throughout the analysis (p = 0.1616, p = 0.2829 and p = 0.2882, respectively).

## Conclusion

There was an immediate clinical reduction of halitosis with aPDT and tongue scraping, but there was no reduction in the number of bacteria throughout the study, or differences in the control times, both in the clinical and microbiological results. New clinical trials are necessary to better assess the tested therapies.

## Trial registration

Clinical Trials NCT03996044.

## Introduction

Halitosis is a multifactorial condition, which may have a local or systemic origin, characterized by the presence of a bad odor in the oral cavity. This condition has a negative impact on the subjects's self-esteem and social life [1]. After caries and periodontal disease, halitosis is the third most common complaint in dental offices [2]. Periodontitis and their biomarkers galectin 3 and NLRP3 are related with halitosis in very recent literature. Current studies have shown that the levels of these substances are higher in the saliva of patients with periodontitis [3,4]. Many other factors are involved in the etiology of halitosis, the main one being the accumulation of bacteria on the back of the tongue [1]. These bacteria mainly produce three volatile sulfur compounds: sulfhydride ($H_2S$), dimethylsulfide (($CH_3$)$_2$S), and methylmercaptan ($CH_3SH$). The production of these compounds is related to the presence of halitosis [5].

There are three categories of halitosis: genuine halitosis, pseudo-halitosis, and halitophobia, the last two being related to psychological conditions [6]. There are also three diagnostic methods: organoleptic test, which consists of a subjective test based on the perception of a previously calibrated evaluator; portable breath tester, which presents results similar to the organoleptic test, but does not present specificity for volatile sulfur compounds; and gas chromatography, which is considered the gold standard in the diagnosis of halitosis [6–8].

Since the etiology is mainly linked to the accumulation of bacteria on the dorsum of the tongue [1], treatments for halitosis aim to reduce the microbial colonization, either by mechanical action with the use of tongue scrapers, chemical action with the use of antiseptic mouthwashes, biological products such as probiotics, or their association [2,9]. Conventional treatments (scrapers and mouthwashes) have several disadvantages such as discomfort for the patient, and pigmentation of the teeth, among others. Therefore, alternative treatments have emerged such as probiotics and antimicrobial photodynamic therapy (aPDT). aPDT involves the use of a light source (light-emitting diodes (LED) or lasers) and a photosensitizer compatible with the light, and the interaction of light with the photosensitizer generates reactive species of oxygen, inducing bacterial cell death [2]. Probiotics are living microbial cells with beneficial effects to health, which may control the resident microbiota [10]. Lactobacillus salivarius are currently used as probiotic for the control of several infectious/inflammatory diseases [11]. The filtrate of a culture of the probiotic strain L. salivarius WB21 is able to inhibit

growth of the proteolytic oral organism Porphyromonas gingivalis [12], and oral administration of L.salivarius WB21 was able to control halitosis, the concentration of exhaled volatile sulfur compounds and improved periodontal clinical parameters in patients with malodor [13,14].

Although both aPDT and the use of probiotics are being researched in the literature for the treatment of halitosis, there is a lack of controlled clinical and microbiological trials in which the therapies are tested and compared. Therefore, we aimed to evaluate the effect of a combination treatment of antimicrobial photodynamic therapy and use of the probiotic L.salivarius WB21 on halitosis assessed by gas chromatography and the levels of proteolytic bacteria determined by quantitative PCR on the tongue coating, in a short term longitudinal study.

## Materials and method

The project was approved by the Ethics Committee of Universidade Nove de Julho, under process number 20123519.4.0000.5511 (CAAE) and feedback number 3669442. It was registered in ClinicalTrials.gov, under the number NCT03996044, first posted on 2019-06-21and last updated on 2023-08-18. This study follows a previously published protocol [7], that was in accordance with SPIRIT (Standard Protocol Items: Recommendations for Interventional Trials) guidelines. This clinical trial follows the CONSORT (Consolidated Standards of Reporting Trials) guidelines.

Fifty-two individuals of both sexes, recruited at the Dental Clinic of Universidade Nove de Julho, were included. Inclusion criteria: participants from 18 to 25 years old, young adults, with a diagnosis of halitosis exhaling sulfhydride (H2S) $\geq$ 112 ppb on gas chromatography, seeing as this is the main gas originated from the tongue coating. Exclusion criteria comprised those with dentofacial anomalies (such as cleft lip, palatine, and nasopalatine fissures), undergoing orthodontic and/or orthopedic treatment, undergoing oncological treatment, with systemic alterations (gastrointestinal, renal, hepatic), undergoing antibiotic treatment up to 1 month before the survey, pregnant women, and those with fissured or furrowed tongue. These participants were excluded mainly because these conditions could alter their microbiota, making their data too discrepant to be compared with participants without them.

### Sample size calculation

The sample size was calculated based on the data reported by Costa da Mota *et al.* [15]. The sample size was estimated using an F test model with 4 groups, a significant level of $\alpha = 0.05$ and an estimated test power of 1- $\beta$ 80%. The initial sample size estimation was 11 subjects per group. However, to account for the possible non-parametric distribution of the data, 15% more subjects were added to each group. group [Erich L. Lehmann, Nonparametrics: Statistical Methods Based on Ranks, Revised, 1998, ISBN = 978–0139977350, pages 76–81.] resulting in 13 subjects per group. The calculation was caried out using G*Power V. 3.1.97.

It is important to note that the sample size estimation is not a fixed number, and it is subject to change based on the results of the study or unexpected events such as dropouts or missing data. The use of G*Power 3.1.9.6 to perform the calculations can help ensure that the study has adequate power to detect a significant effect size and reduce the risk of a type II error.

The 52 participants were instructed, through classes and files that were sent via WhatsApp, to brush with a toothpaste containing amine fluoride (Elmex®) and to use dental floss 3 times a day after meals, for 30 days. Then, participants were randomly allocated into 4 experimental groups (n = 13 per group). Group 1: recommended to use a tongue scraper before bedtime, daily; Group 2: treated with one session of aPDT with Bixa orellana extract and blue LED, applied to the back and middle thirds of the tongue; Group 3: Instructed to ingest capsules

containing Lactobacillus salivarius WB21 (6.7 x 108 CFU) and xylitol (280mg), 3 times a day after meals, for 14 days; Group 4: treated with aPDT and probiotics.

Patients were randomized by block randomization into the groups (www.randomizer.at), according to the treatment to be performed. Opaque envelopes were identified with each number and inside it a sheet containing the information of the corresponding experimental group was inserted. The envelopes were opened in order, showing the group to which, the participant would belong to when they were included. The envelopes were sealed and remained sealed in numerical order in a safe place until the procedures were made. The person in charge of the drawing process was not involved in the other parts of the study. The researchers responsible for the procedures were the ones to enroll participants and assign them to interventions. There were no exclusions after randomization. Recruitment started to take place in January 20, 2021, ended in June 30, 2021, and the follow-ups were completed by November 2021. Written consent was obtained from all participants, they signed an Informed Consent Form, that was previously approved by the ethics committee.

The activity flowchart is represented in Fig 1.

Outcome assessments performed before and after the treatments employed

For all groups, firstly, the participants were instructed, through a lecture and files, to brush with a toothpaste containing amine fluoride in its composition (Elmex®) and to use dental

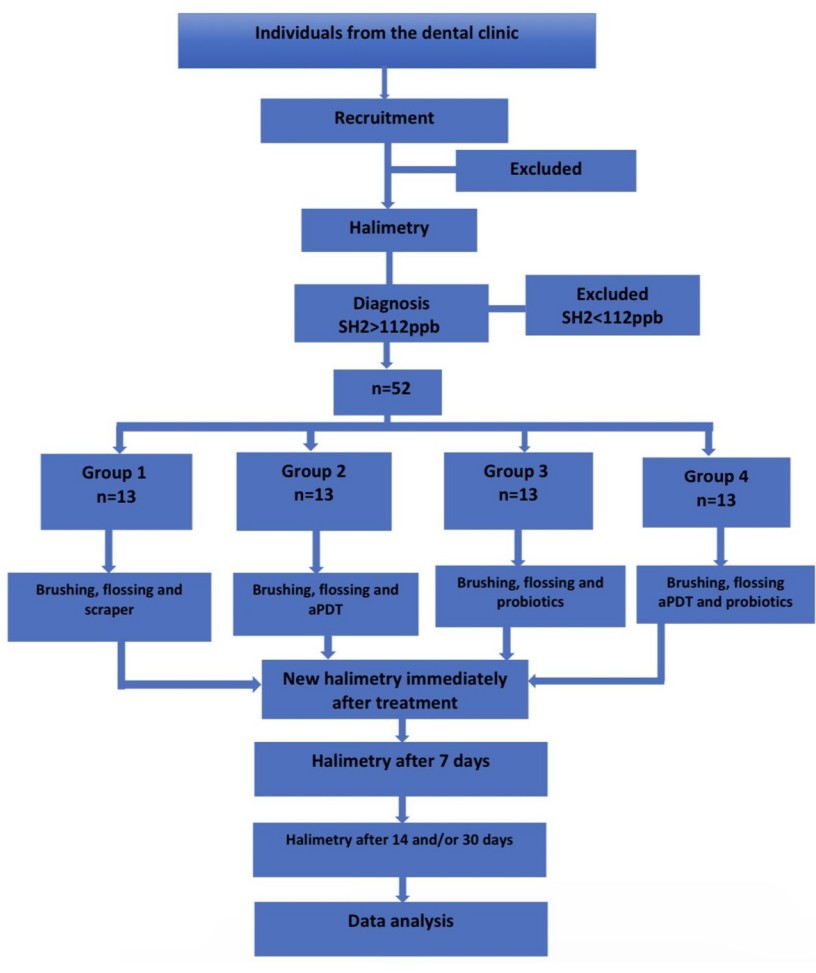

**Fig 1. Activity flowchart.**

floss, 3 times a day after meals for 30 days. The technique taught to the participants was that of Bass, in which the bristles of the brush are positioned at an angle of approximately 45o towards the inside of the gingival sulcus, both on the free and proximal surfaces, in addition to short and slightly circular vibrating movements [16,17]. Then, an initial assessment of tongue coating was performed using the Coated Tongue Index (CTI) proposed by Shimizu et al. [18]. The tongue was divided into 9 parts and for each part, a score was given, being 0 –absence of tongue coating, 1 –presence of tongue coating with visible papillae, 2 –thickness of tongue coating that makes it impossible to visualize the papillae. These grades were added, divided by 18, and multiplied by 100, to obtain a final index of 0–100%. It should be clarified that the participants were only instructed and did not brush and floss in the same treatment session. Afterward, the evaluation was made by gas chromatography with the OralChromaTM device and the microbiological collection for later evaluation by PCRq in real time.

The collection of oral air followed the manufacturer's instructions (2 Oral ChromaTM Manual Instruction), in which the participant was instructed to rinse with cysteine (10 mM) for 1 minute, then remain with their mouth closed for another 1 minute. A syringe from the same manufacturer designed to collect oral air was introduced into the patient's mouth. For 30 seconds, the patients remained with their mouths closed, breathing through the nose, without touching the syringe with the tongue. The plunger was pulled out, the syringe was again emptied of air into the patient's mouth, and the plunger was pulled again to fill the syringe with the breath sample. The tip of the syringe was wiped with gauze to remove moisture from the saliva, the gas injection needle was placed on the syringe, and the plunger was adjusted to 0.5 ml. The collected gases were injected into the inlet port of the device with a single movement. Sulfhydride is a gas whose origin is mainly from the bacteria present on the back of the tongue. Values above 112 ppb are indicators of halitosis, so participants who presented these values were included in the study. To avoid changes in halimetry, participants were instructed to follow these guidelines: 48 hours before the assessment, avoid eating foods with garlic, onions, and strong spices, alcohol consumption and use mouthwash. On the day of the assessment, in the morning, they could eat up to a maximum of 2 hours before the exam, abstain from coffee, candies, chewing gum, and oral and personal hygiene products with perfume (aftershave, deodorant, perfume, creams and/or tonic) and brushing should only be done with water [19–24]. The halimetry process with OralChromaTM was performed before, immediately after the treatments, 7 days, 14 days, and 30 days after the initial collection, depending on the groups. In Group 1 (scraper) and in Group 2 (aPDT), the measurements were taken at the initial times, immediately after, 7 days, and 30 days, for control. In Group 3 (probiotics), the initial times, 7 days, 14 days, and 30 days were performed. In this group, it was not possible to carry out the "immediately after" time, since the participant had to start ingesting the probiotics. Consequently, the 14-day control period was added, as the participant ingested the capsules for 14 days. In Group 4 (aPDT + probiotics), all 14 times were performed (initial, immediately after, 7 days, 14 days, and 30 days after).

Microbiological samples were collected from the dorsum of the tongue before and after aPDT, as well as after 7, 14, and 30 days. A swab was used to collect the samples that were placed in sterile tubes containing Tris–EDTA. Then, they were docketed and stored until analysis at– 80˚C. After defrosting, the samples were shaken at the vortex for 1 min. The Master Pure DNA Extraction Kit (Epicentre Technologies Corp., Chicago, IL, USA) was used to DNA extraction, following the manufacturer's instructions. Purified DNA was re-suspended in TE buffer. Through real-time PCRq, Porphyromonas gingivalis was investigated and quantified in the samples. The species-specific 16S rRNA gene for P. gingivalis was previously cloned into a plasmid and inserted into Escherichia coli DH5-alpha, the quantitation in each sample was compared to a standard curve with this product [25]. The curve, which covered 10 and 108

plasmids, was compared with the DNA of the samples, this process was performed after the serial dilutions. Four copies of the 16S rRNA gene/chromosome were used to calculate the number of P. gingivalis cells [18 Negative control was performed with Sterile Milli Q Water instead of mold DNA. To perform the quantitative PCR, a Step One Plus Real-Time PCR System (Applied Biosystems, Foster City, CA, USA) was used, and fluorescence was used to detect the products, using SYBR® Green PCR Master Mix (Applied Biosystems, Thermo Fisher Scientific), following the protocol recommended by the manufacturer. Standard curve data in triplicate were compared with sample results to perform quantitative analysis.

For the quantification of 16S rRNA from P. gingivalis in the tongue coating samples, species specific primers were used: 5′-TGT AGA TGA CTG ATG GTG AAA ACC-3′ and 5′-ACG TCA TCC CCA CCT TCC TC-3′ [26]. The reaction was performed with a total volume of 10 μL, containing 5 μL of Syber Green, 2 μL of sample, and 200 mM of each primer for 16S rRNA. The thermocycling protocol was the following: 95˚C/10 min, 20 cycles at 95˚C/15 s, 60˚C/1 min, and 83˚C/10 s, a melting curve of 95˚C/15 s, 70˚C/1 min (+ 0.6˚C), and 95˚C/15 s.

We only considered reactions in which the efficiency was of 100% (± 10%), with R2 close to 1.

For the quantification of 16S rRNA from Treponema denticola in the tongue coating samples, species-specific primers were used: 5′ CGTTCCTGGGCCTTGTACA3′ and 5′ TAGCGACTT CAGGTACCCTCG3′ [27].

The reaction was performed with a total volume of 10 μL, containing 5 μL of Syber Green, 2 μL of sample, and 200 mM of each primer for 16S rRNA. The thermocycling protocol was 95˚C/10 min, 40 cycles at 95˚C/15 s, 60˚C/1 min, and 83˚C/10 s, a melting curve of 95˚C/15 s, 60˚C/1 min (+ 0.6˚C), and 95˚C/15 s. We only considered reactions in which the efficiency was of 100% (± 10%), with R2 close to 1.

For the quantification of 16S rRNA from Tannerella forsythia the tongue coating samples, species-specific primers were used (5′ GGGTGAGTAACGCGTATGTAACCT3′ and 5′ ACCCATC CGCAACCAATAAA3′) [28]. The reaction was performed with a total volume of 10 μL, containing 5 μL of Syber Green, 2 μL of sample, and 200 mM of each primer for 16S rRNA. The thermocycling protocol was 95˚C/10 min, 40 cycles at 95˚C/15 s, 60˚C/1 min, and 74˚C/10 s, a melting curve of 95˚C/15 s, 60˚C/1 min (+ 0.6˚C), and 95˚C/15 s. We only considered reactions in which the efficiency was of 100% (± 10%), with R2 close to 1.

## Treatment groups

**Group 1–Tongue scraping.** In 13 individuals, tongue scraping was performed by the same operator in all participants. Posteroanterior movements were performed with the scraper on the tongue dorsum, followed by cleaning the scraper with gauze. This procedure was performed ten times in each patient, with the aim of standardizing the mechanical removal of the tongue coating.

**Group 2-Antimicrobial photodynamic therapy (aPDT).** In the other 13 individuals, the LED light curing device–Valo Cordless Ultradent®, an office device, with coupled radiometer, a spectrum of 440-480nm, 2 and irradiance of 450mW/cm was used. One session of aPDT was performed with the photosensitizer (PS) Bixa orellana extract manipulated at a concentration of 20% (Formula e Ação®), in spray, applied in sufficient quantity to cover the middle third and back of the tongue (5 sprays) for 2 minutes for incubation. The excess was removed with a sucker in order to keep the surface moist with the PS itself, without using water. Six points were irradiated with a distance of 1 cm between points, considering the halo of light scattering and the effectiveness of aPDT. The apparatus was previously calibrated with a wavelength of 395–480 nm, for 20 seconds per point, and energy of 9.6J, and the light was irradiated so that a halo of 2 cm in diameter per point was formed [22,23].

**Group 3-Probiotics.** The 13 participants in this group were instructed to ingest probiotic capsules. Pharmacy compounded capsules containing strains of Lactobacillus salivarius WB21 (6.7 x 108 CFU) and xylitol (280mg) were used. Forty-two capsules were given to each patient, who had to take 1 capsule, 3 times a day after meals, for 14 days.

**Group 4-Antimicrobial photodynamic therapy (aPDT) and probiotics.** The 13 participants in this group received both treatments described in Groups 2 and 3.

## Results

### Statistical analysis

The data were analyzed for their normality using the Shapiro-Wilk test. As the data showed no normality, they were compared using the Kruskal-Wallis test (independent measures) and Friedman ANOVA (dependent measures) followed by appropriate posthoc tests, when necessary (Dunn's test for independent measurements and pairwise Wilcoxon signed-rank test with Ryan-Holm stepdown Bonferroni procedure for adjustment). The significance level was $\alpha = 0.05$. For this work we used SPSS V. 25.0 for the inferential analysis.

Table 1 shows the data of gender, age, and Coated Tongue Index (CTI) of the research subjects.

The between-group analysis was performed for each time studied individually. At the initial time, there was no significant difference between the studied groups ($p = 0.0706$, Kruskal-Wallis ANOVA), indicating that the groups were well-balanced in relation to the initial condition. The analysis of the other times after the 15 treatments showed little difference between the studied groups ($p = 0.9581$, $p = 0.6187$, and $p = 0.9635$ for the times "immediately", "7 days" and "30 days", respectively. Kruskal -Wallis ANOVA). The analysis of the "Scraper" group showed that the time "immediately after" differs significantly from all other times ($p = 0.0006$, Friedman), as can be seen in Fig 2.

The analysis of the "aPDT" group showed that the time "immediately after" differs significantly from all other times ($p = 0.0008$, Friedman), as can be seen in Fig 3.

The analysis of the "Probiotics" group showed no significant difference in the times studied ($p = 0.7530$, Friedman), as can be seen in Fig 4.

The analysis of the "aPDT+Probiotic" group showed that the time "immediately after" differs significantly from the time "30 days" ($p = 0.0008$, Friedman), as can be seen in Fig 5.

### Microbiological results

Analysis by qPCR was performed in all analyzed patients. After verifying the lack of normality of the data using the Kolmogorov-Smirnov test, the Kruskal-Wallis rank sum test was performed with Dunn's post-test.

Firstly, the data from the analysis by qPCR for the bacteria Porphyromonas gingivalis, Tannerella forsythia, and Treponema denticola (Figs 6–8) is presented. The results of the quantitative analysis of each of the bacteria were represented by their means and standard errors.

**Table 1. Demographic data of research subjects.**

|  | Scraper (n = 13) | aPDT (n = 13) | Probiotics (n = 13) | aPDT + Probiotics (n = 13) |
|---|---|---|---|---|
| **Men** | 1 (8%) | 3(23%) | 3 (15%) | 3 (23%) |
| **Women** | 12 (92%) | 10 (77%) | 11 (85%) | 10 (77%) |
| **Age (years ±SD)** | 21,0 ± 1,1 | 21,3 ± 1,9 | 21.3 ± 1,9 | 21.3 ± 1.7 |
| **CTI ±SD** | 25% ± 14% | 32% ± 17% | 48% ± 19% | 42% ± 19% |

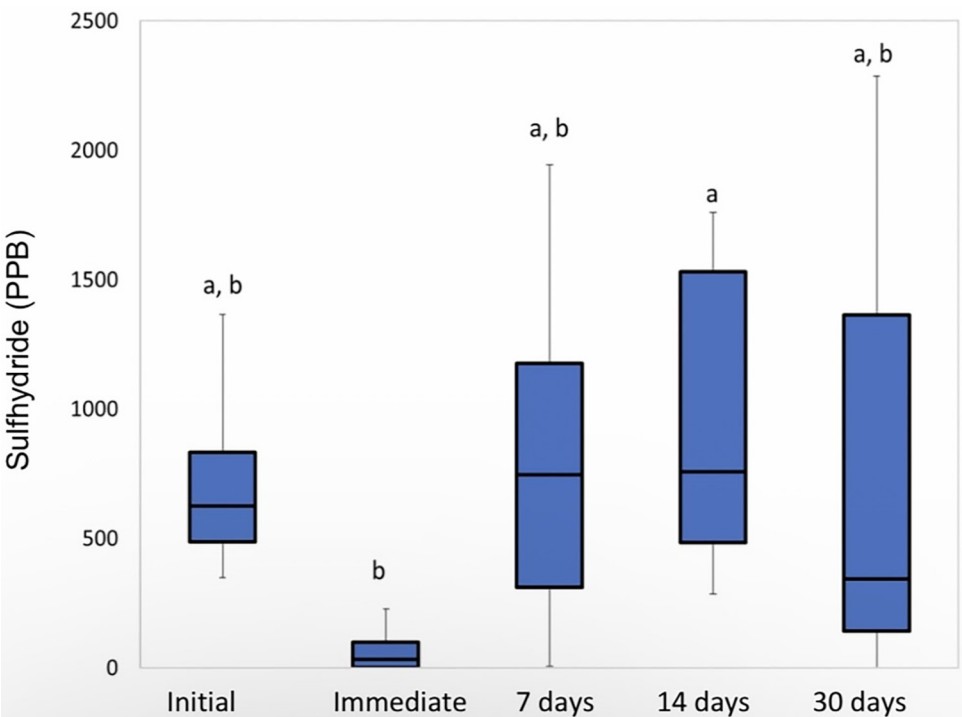

**Fig 2. Sulfhydride analysis for the "Scraper" group.** Different letters mean significant difference.

In the time domain, it was observed that there was no statistically significant difference (p = 0.1616, Kruskal-Wallis) for any of the groups studied for the bacteria Porphyromonas gingivalis. Therefore, the observed differences are due to random fluctuations inherent to the experiment.

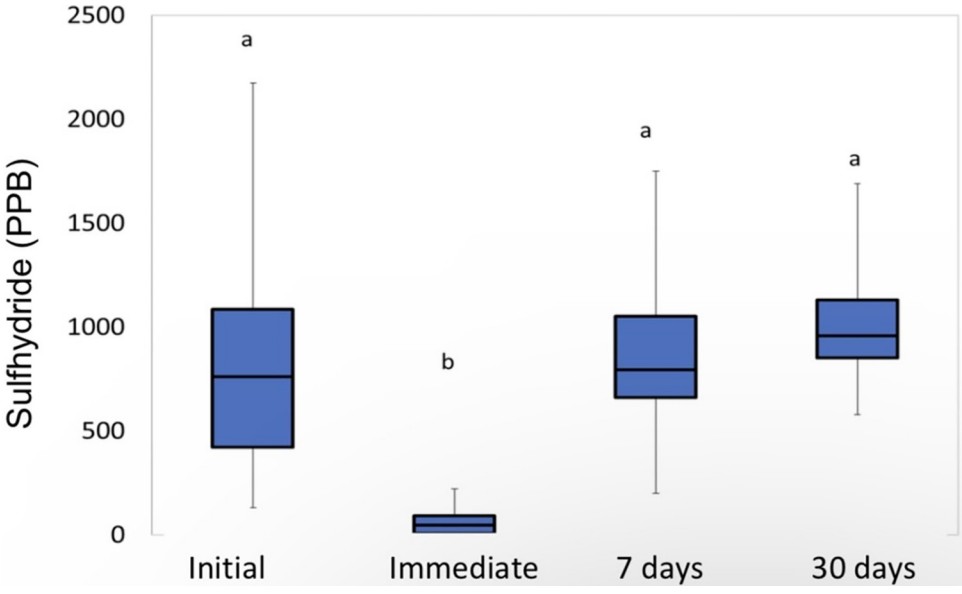

**Fig 3. Sulfhydride analysis for the "aPDT" group.** Different letters mean significant difference.

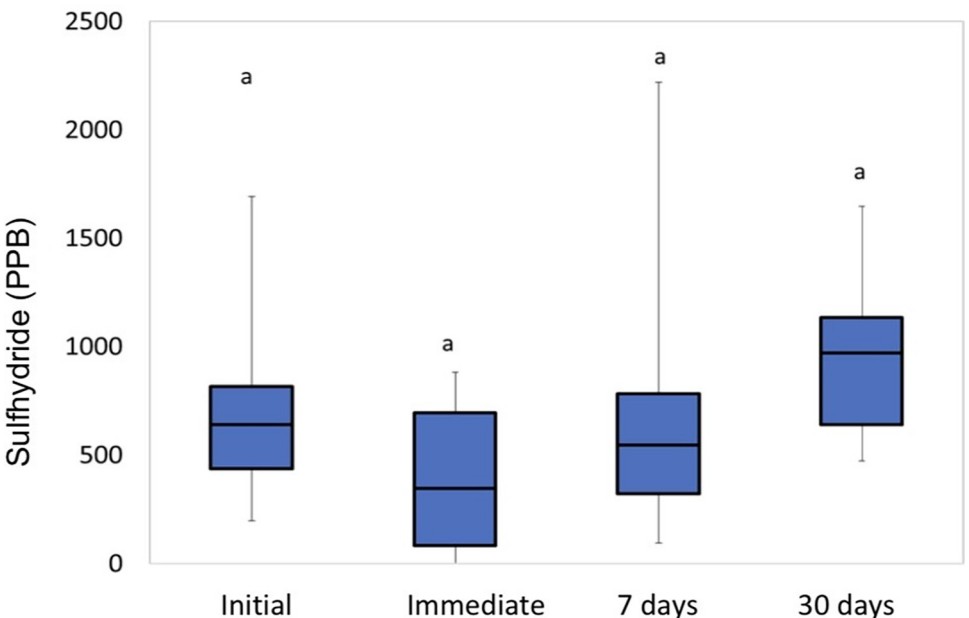

**Fig 4. Sulfhydride analysis for the "Probiotics" group.** Different letters mean significant difference.

For Tannerella forsythia, no difference was observed between the groups in the analyzed times (p = 0.2829, Kruskal-Wallis). In the time domain, it was also observed that there was no statistically significant difference for any of the studied groups. No difference was found for Treponema denticola, as well (p = 0.2882, Kruskal-Wallis).

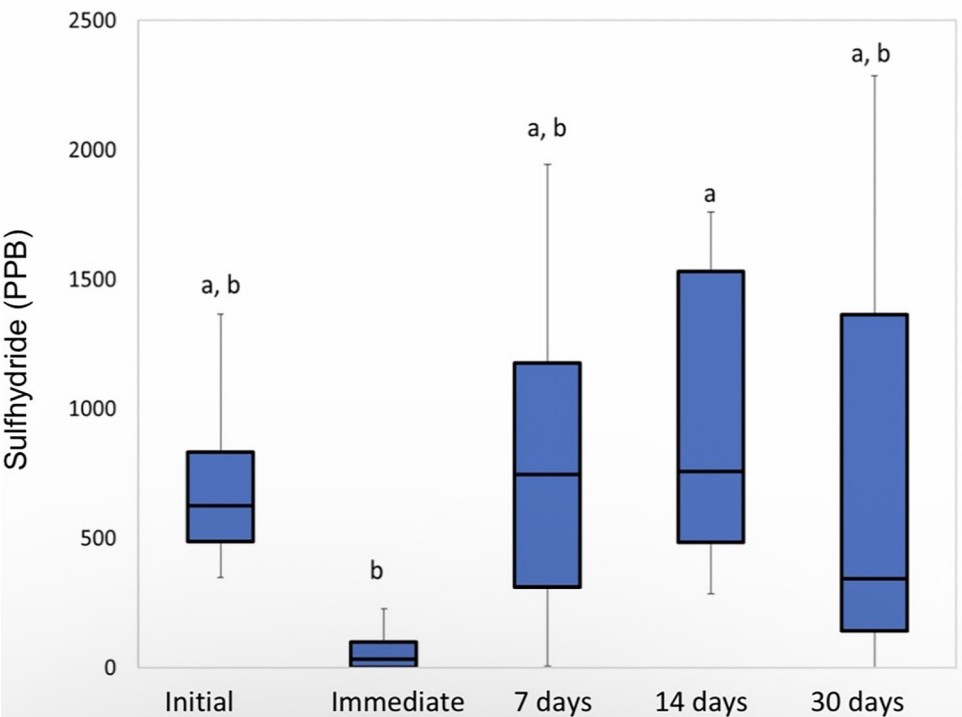

**Fig 5. Sulfhydride analysis for the "PDT + Probiotic" group.** Different letters mean significant difference.

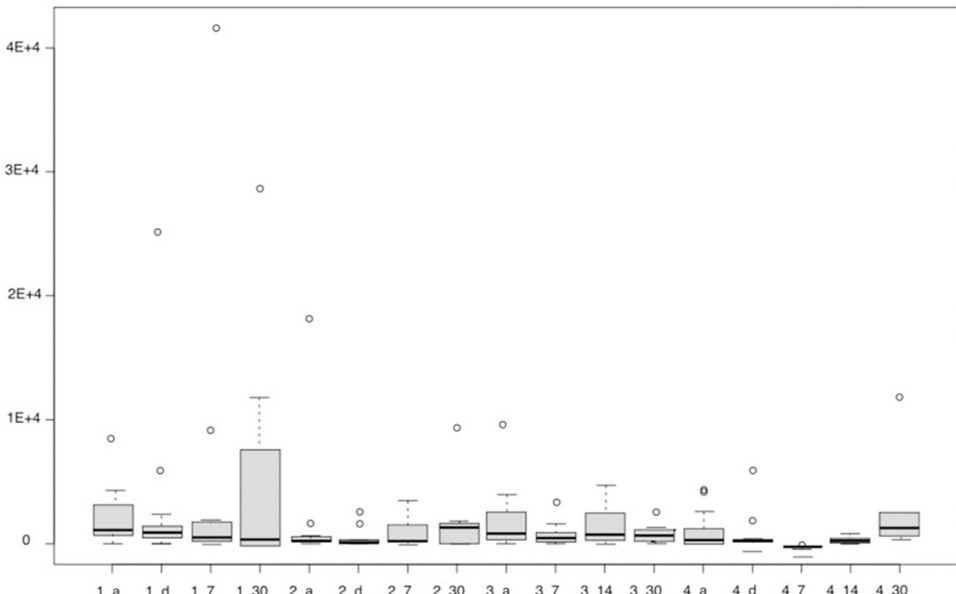

**Fig 6. The figure represents the groups in each proposed period for Porphyromonas gingivalis.** Group 1—Scraper (a-initial, d- immediately after, 7 and 30 days after), Group 2—aPDT (a-initial, d- immediately after, 7 and 30 days after), Group 3 –Probiotics (a-initial, 7 and 30 days after), Group 4—aPDT + probiotics (a-initial, d-immediately after, 7, 14 and 30 days after).

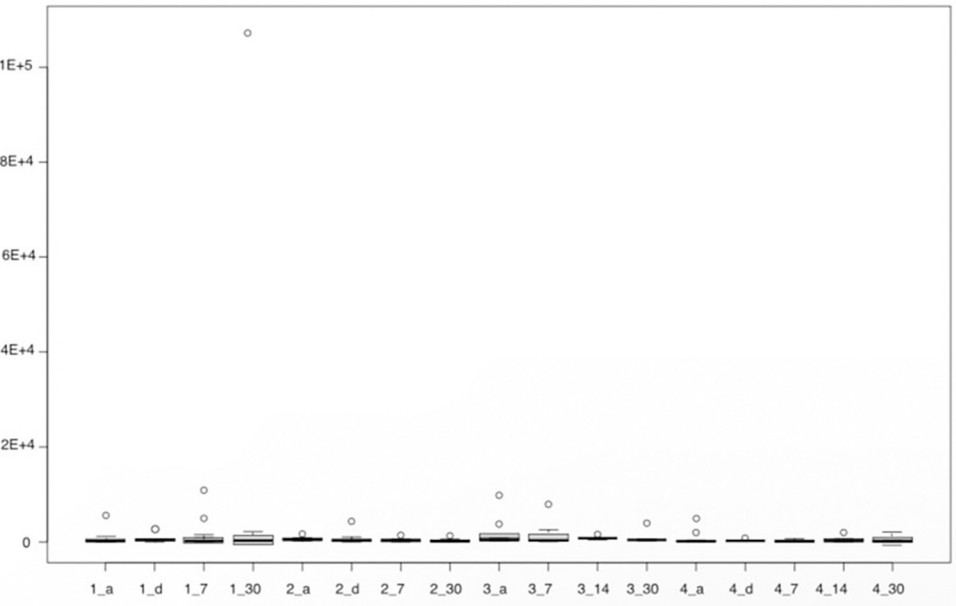

**Fig 7. The figure represents the groups in each proposed period for Tannerella forsythia.** Group 1—Scraper (a-initial, d- immediately after, 7 and 30 days after), Group 2—aPDT (a-initial, d- immediately after, 7 and 30 days after), Group 3 –Probiotics (a-initial, 7 and 30 days after), Group 4—aPDT + probiotics (a-initial, d-immediately after, 7, 14 and 30 days after).

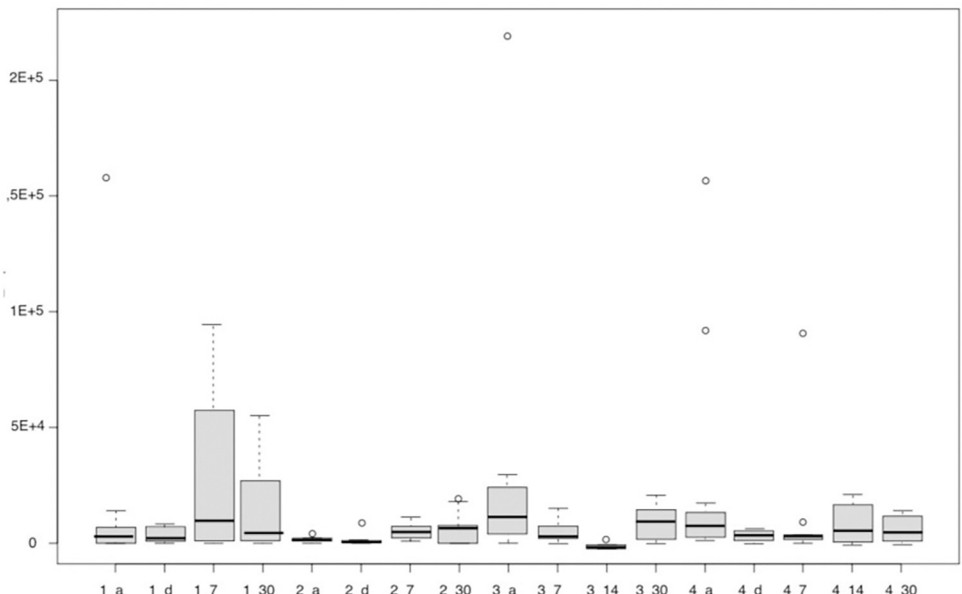

**Fig 8. The figure represents the groups in each proposed period for Treponema denticola.** Group 1—Scraper (a-initial, d- immediately after, 7 and 30 days after), Group 2—aPDT (a-initial, d- immediately after, 7 and 30 days after), Group 3 –probiotics (a-initial, 7 and 30 days after), Group 4—aPDT + probiotics (a-initial, d- immediately after, 7, 14 and 30 days after).

## Discussion

The results of the present study showed that there was an immediate reduction in halitosis with aPDT and/or scraper in the clinical evaluation (halimetry). However, probiotics showed no difference in relation to the initial levels. The amount of *Porphyromonas gingivalis*, *Tannerella forsythia* and *Treponema denticola* were not altered throughout the analysis.

The clinical results obtained so far have shown that the treatments had an immediate effect in reducing halitosis. These results corroborate articles in which photodynamic therapy was used to treat this condition. In the article by Mota et al., 2021 [24], methylene blue was used in association with a red LED. In this study, aPDT, tongue scraping, and the two treatments combined were used. All of these modalities showed an immediate reduction in halitosis, but the effect was maintained after 7 days only in the tongue-scraping group. This difference did not occur in the current study, differing in this particular result. In the microbiological analysis by Mota et al., 2021 [24], no statistically significant differences were found in the analyzed groups/bacteria. They concluded that aPDT with 0.005% methylene red and blue LED caused an immediate reduction in halitosis, but the effect was not maintained after 7, 14, or 30 days, with results similar to those obtained clinically in this project. There was no reduction in the number of bacteria investigated or in the quantification of universal 16S rRNA, neither in our study nor in theirs. Bixa orellana extract with blue LED was also used for aPDT on the tongue dorsum in the study by Gonçalves et al., 2020 [23]. In this study, patients received treatment with aPDT, tongue scraper, or both modalities combined. In all groups, there was a difference between the baseline value and the value immediately after treatment, with similar results to those obtained in this protocol. However, in the seven-day control carried out in the article, the levels of sulfhydride returned to initial levels, showing that the reduction was not maintained during this period. Only clinical results were reported in the study in question, and the results corroborate the findings of the current trial.

The immediate reduction of sulfhydride levels with aPDT has already been verified in the literature [19–21,23,24]. In contrast, the systematic review carried out by Motta et al., 2022 [29], shows that the effects of aPDT on halitosis are uncertain and that additional clinical trials, with a larger number of participants and long-term evaluations, such as the present study, need to be performed to support the use of this intervention. The issue of control periods really remains dubious. Therefore, in this project, more measurements were carried out.

Moreover, unlike the previous studies already mentioned, in this study, hygiene instructions, guidance on brushing, flossing, and how to use the lingual scraper (in the Scraper group) were given. We believed that the association of correct hygiene habits with the proposed treatments would be effective in maintaining immediate results. However, such results were not found in control assessments. On the other hand, it is important to remember that periodontitis and its markers are also related to halitosis in the literature. A study showed that patients with periodontitis and periodontitis and diabetes exhibited high serum and salivary NLRP3 inflammasome levels compared to diabetes patients and healthy subjects. Periodontitis and HDL-cholesterol were significant predictors of serum NLRP3 while periodontitis and CRP were the significant predictors of salivary NLRP3 [3]. Another article showed that patients with periodontitis and periodontitis + CHD presented significant higher serum and salivary Galectin-3 levels in comparison with CHD patients and healthy subjects. Periodontitis and ET-1 were the significant predictors of serum and salivary Galectin-3 levels, respectively [4].

Regarding the use of probiotics, this type of treatment has already been used in other clinical trials to reduce bacteria in the oral cavity. A systematic review was performed in 2019 [30], to summarize the evidence on the effect of probiotics on halitosis. Clinical trials with any type of probiotic treatment for at least 2 weeks, the protocol followed in our study (participants took the probiotics for 14 days), were included. Meta-analysis revealed that organoleptic assessment scores were significantly lower in subjects receiving probiotics than in placebo groups, but no significant difference was observed in VSC concentration, results similar to our sulfhydride analysis. Another systematic review [31], carried out in 2022, pointed out that the Lactobacillus species, also used in this project, is the most proposed one for the treatment of halitosis. Both reviews agree that the available evidence is insufficient for probiotic recommendations for oral malodor. More clinical trials, like this one, should aim for longer follow-up and standardized administration methods to prove or disprove the effect of probiotics on halitosis. Coherently with these recommendations, in the present study, the times of 14 and 30 days for reassessment were carried out, as well as the standardization of the type of probiotic to be ingested and the guidelines for the participants, which were given by the same researcher.

In the recent past, the most studied bacteria in patients with halitosis were the Gram-negative Fusobacterium nucleatum, Porphryomonas gingivalis, Treponema denticola, and Tannerella forsythia [32–38]. For this reason, one of our objectives was to quantify these bacteria on the dorsum of the tongue.

However, in the quantitative analysis by qPCR, it was observed that there was no difference (p>0.05) for any of the groups analyzed over time.

As study limitations, we can cite the lack of control the researchers had over the consumption of the probiotics, seeing as the participants took them at home and may not have followed instructions correctly. Same goes to oral hygiene instructions, which were participant dependent.

## Conclusions

Our clinical results showed an immediate reduction in halitosis with aPDT, with the use of the scraper, and with the use of the two therapies combined. The immediate result shows the

effectiveness of the treatments, for a period shorter than that of 7 days, which is yet to be discovered. The use of probiotics showed no difference in the levels of sulfhydride, in relation to the initial levels. The purpose of the seven-day control period for all groups was to evaluate the results of the therapies after one week, simulating control of patients in clinical dental practice, and verifying the possible maintenance of treatments. The 14-day period was carried out in the groups that took the probiotics, to check the result at the end of the ingestion period. 30-days controls were performed in all groups to verify possible long-term treatment changes. The controls showed a reduction in sulfhydride levels only in the collection immediately after the therapies. As for the microbiological results, when the tongue coating samples were analyzed by quantitative PCR for Porphyromonas gingivalis, Tannerella forsythia, and Treponema denticola, it was not possible to observe differences between the groups for any of the evaluated bacteria. New clinical trials with different treatment parameters are needed to verify the possible effectiveness of aPDT and the use of probiotics in reducing halitosis.

## Supporting information

**S1 Checklist. Filled consort checklist statement.**
(DOC)

**S1 File. Protocol in original language.** Protocol for the clinical trial in original language.
(DOCX)

**S2 File. Protocol.** Protocol published in English.
(DOCX)

**S1 Data. Raw research data.**
(ZIP)

## Author Contributions

**Conceptualization:** Pamella de Barros Motta, Marcela Leticia Leal Gonçalves, Ana Paula Taboada Sobral, Lara Jansiski Motta, Anna Carolina Ratto Tempestini Horliana.

**Data curation:** Marcela Leticia Leal Gonçalves, Juliana Maria Altavista Sagretti Gallo, Ana Paula Taboada Sobral, Anna Carolina Ratto Tempestini Horliana.

**Formal analysis:** Juliana Maria Altavista Sagretti Gallo, Ana Paula Taboada Sobral, Lara Jansiski Motta, Dione Kawamoto, Alessandro Melo Deana, Anna Carolina Ratto Tempestini Horliana.

**Funding acquisition:** Pamella de Barros Motta, Sandra Kalil Bussadori.

**Investigation:** Marcela Leticia Leal Gonçalves, Juliana Maria Altavista Sagretti Gallo, Ana Paula Taboada Sobral, Anna Carolina Ratto Tempestini Horliana.

**Methodology:** Marcia Pinto Alves Mayer, Anna Carolina Ratto Tempestini Horliana.

**Project administration:** Lara Jansiski Motta, David Casimiro de Andrade, Kristianne Porta Santos Fernandes, Anna Carolina Ratto Tempestini Horliana.

**Resources:** Kristianne Porta Santos Fernandes, Sandra Kalil Bussadori.

**Software:** Lara Jansiski Motta, Dione Kawamoto, Kristianne Porta Santos Fernandes.

**Supervision:** Lara Jansiski Motta, Marcia Pinto Alves Mayer, Dione Kawamoto, David Casimiro de Andrade, Elaine Marcílio Santos, Kristianne Porta Santos Fernandes, Raquel Agnelli Mesquita-Ferrari, Sandra Kalil Bussadori.

**Validation:** Ana Paula Taboada Sobral, Lara Jansiski Motta, Elaine Marcílio Santos, Kristianne Porta Santos Fernandes, Raquel Agnelli Mesquita-Ferrari.

**Visualization:** Marcia Pinto Alves Mayer, Elaine Marcílio Santos, Kristianne Porta Santos Fernandes, Raquel Agnelli Mesquita-Ferrari, Sandra Kalil Bussadori.

**Writing – original draft:** Pamella de Barros Motta.

**Writing – review & editing:** Marcela Leticia Leal Gonçalves, Marcia Pinto Alves Mayer, David Casimiro de Andrade, Sandra Kalil Bussadori.

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
