## [Decision Letter · Decision Letter 0]

20 Sep 2023

PONE-D-23-25648Short term effect of antimicrobial photodynamic therapy and probiotic L. salivarius WB21 on halitosis: a controlled and randomized clinical trialPLOS ONE

Dear Dr. Kalil Bussadori,

Thank you for submitting your manuscript to PLOS ONE. After careful consideration, we feel that it has merit but does not fully meet PLOS ONE’s publication criteria as it currently stands. Therefore, we invite you to submit a revised version of the manuscript that addresses the points raised during the review process.

We look forward to receiving your revised manuscript.

Kind regards,

Gaetano Isola, Ph.D.

Academic Editor

PLOS ONE

Journal Requirements:

-https://doi.org/10.1007/s12602-017-9351-1 

In your revision ensure you cite all your sources (including your own works), and quote or rephrase any duplicated text outside the methods section. Further consideration is dependent on these concerns being addressed.

5. We note that the original protocol that you have uploaded as a Supporting Information file contains an institutional logo. As this logo is likely copyrighted, we ask that you please remove it from this file and upload an updated version upon resubmission.

Reviewers' comments:

Reviewer's Responses to Questions

**Comments to the Author**

1. Is the manuscript technically sound, and do the data support the conclusions?

Reviewer #1: Yes

Reviewer #2: No

2. Has the statistical analysis been performed appropriately and rigorously? 

Reviewer #1: Yes

Reviewer #2: No

3. Have the authors made all data underlying the findings in their manuscript fully available?

Reviewer #1: Yes

Reviewer #2: Yes

4. Is the manuscript presented in an intelligible fashion and written in standard English?

Reviewer #1: No

Reviewer #2: No

5. Review Comments to the Author

Reviewer #1: In the manuscript entitled: "Short term effect of antimicrobial photodynamic therapy and probiotic L. salivarius WB21 on halitosis: a controlled and randomized clinical trial" the authors analyzed effect of 4 antimicrobial photodynamic therapy (aPDT) and the use of probiotics on halitosis.

The authors found that there was an immediate clinical result, but no reduction in the 50 number of bacteria.

Major comments:

In general, the idea and innovation of this study regards the analysis of antimicrobial photodynamic therapy and probiotic L. salivariuis interesting and novel because the role these aspects in medicine are validated but further studies on this topic could be an innovative issue in this field could be open a creative matter of debate in literature by adding new information. Moreover, there are few reports in the literature that studied this interesting topic with this kind of study design.

The study was well conducted by the authors; However, there are some concerns to revise that are described below.

The introduction section resumes the existing knowledge regarding the important factor linked with the related factor that influences halitosis.

However, as the importance of the topic, the reviewer strongly recommends, before a further re-evaluation of the manuscript, to update the literature through read, discuss and must cites in the references with great attention all of those recent interesting articles, that helps the authors to better introduce and discuss the role of periodontitis and their biomarjers galectin 3 and NLRP3 related with halitosis 1) oi: 10.1111/jre.12860. PMID: 336411612) doi: 10.1002/JPER.21-0049. PMID: 34008185

The authors should be better specified, at the end of the introduction section, the rationale of the study and the aim of the study. In the central section, should better clarify inclusions and exclusions criteria of the selected patient, with the inclusion/exclusion criteria.

The discussion section appears well organized with the relevant paper that support the conclusions, even if the authors should better discuss the relationship regarding the role exerted by NLRP3 and galectin-3 on both halitosis and related periodontitis progression. The conclusion should reinforce in light of the discussions.

In conclusion, I am sure that the authors are fine clinicians who achieve very nice results with their adopted protocol. However, this study, in my view does not in its current form satisfy a very high scientific requirement for publication in this journal and requests a revision before a futher re-evaluation of the manuscript.

Minor Comments:

Abstract:

-Better formulate the abstract section by better describing the aim of the study

Introduction:

-Please refer to major comments

Discussion

-Please add a specific sentence that clarifies the results obtained in the first part of the discussion

Reviewer #2: Participants diagnosed with halitosis exhaling sulfhydride >= 112 ppb were randomized to one of 4 treatments. The results are unclear.

Major revisions:

1- Abstract: Write the abstract in complete flowing sentences. Clarify the objective of the study. Briefly name the statistical testing methods applied and state the p-values that support the conclusions.

2- Line 115: There appears to be an error with the formatting of one of the dates.

3- Line 128: State and justify the study’s target sample size with a pre-study statistical power calculation. The power calculation should include: (1) the estimated outcomes in each group; (2) the α (type I) error level; (3) the statistical power (or the β (type II) error level); (4) the target sample sizes and (5) the statistical testing method (6) for continuous outcomes, the standard deviation of the measurements.

4- Lines 150-9: Clarify the randomization process. Be more specific and less wordy.

5- Line 297: The data rather than the groups were tested for normality using the Shapiro-Wilk test.

6- Line 297: The following sentence is unclear. What variables were tested using the Kruskal-Wallis or ANOVA tests? An underlying assumption of the ANOVA is normality of data. “As the data showed no normality, they were treated using the Kruskal-Wallis ANOVA test (independent measures) and Friedman ANOVA (dependent measures) followed by appropriate posthoc tests, when necessary.” State the statistical methods used for the posthoc tests.

7- For data that has been collected repeatedly, use a repeated measures statistical approach for analyzing the results.

8- Include a comprehensive and specific statistical analysis section, listing and detailing all the statistical methods used.

9- Express p-values more precisely than p > 0.05.

10- Cite the statistical software used for the analysis.

11- Thoroughly proofread the manuscript.

6. PLOS authors have the option to publish the peer review history of their article (what does this mean?). If published, this will include your full peer review and any attached files.

Reviewer #1: No

Reviewer #2: No

---

## [Author Response · Author response to Decision Letter 0]

4 Nov 2023

Answers for reviewers

Reviewer #1: In the manuscript entitled: "Short term effect of antimicrobial photodynamic therapy and probiotic L. salivarius WB21 on halitosis: a controlled and randomized clinical trial" the authors analyzed effect of 4 antimicrobial photodynamic therapy (aPDT) and the use of probiotics on halitosis.

The authors found that there was an immediate clinical result, but no reduction in the 50 number of bacteria.

Major comments:

In general, the idea and innovation of this study regards the analysis of antimicrobial photodynamic therapy and probiotic L. salivariuis interesting and novel because the role these aspects in medicine are validated but further studies on this topic could be an innovative issue in this field could be open a creative matter of debate in literature by adding new information. Moreover, there are few reports in the literature that studied this interesting topic with this kind of study design.

The study was well conducted by the authors; However, there are some concerns to revise that are described below.

Answer: Firstly, we would like to thank the reviewer for all the comments and suggestions, which will surely help to improve our article. 

Question: The introduction section resumes the existing knowledge regarding the important factor linked with the related factor that influences halitosis.

However, as the importance of the topic, the reviewer strongly recommends, before a further re-evaluation of the manuscript, to update the literature through read, discuss and must cites in the references with great attention all of those recent interesting articles, that helps the authors to better introduce and discuss the role of periodontitis and their biomarjers galectin 3 and NLRP3 related with halitosis 1) oi: 10.1111/jre.12860. PMID: 336411612) doi: 10.1002/JPER.21-0049. PMID: 34008185

Answer: Thank you for your suggestion. Both recommended references were added and highlighted in the text, as follows: “Periodontitis and their biomarkers galectin 3 and NLRP3 are related with halitosis in very recent literature. Current studies have shown that the levels of these substances are higher in the saliva of patients with periodontitis. [3,4]”

Question: The authors should be better specified, at the end of the introduction section, the rationale of the study and the aim of the study. 

Answer: Thank you for your recommendation. We have added and highlighted the information in the text, as follows: “Although both aPDT and the use of probiotics are being researched in the literature for the treatment of halitosis, there is a lack of controlled clinical and microbiological trials in which the therapies are tested and compared. Therefore, we aimed to evaluate the effect of a combination treatment of antimicrobial photodynamic therapy and use of the probiotic L.salivarius WB21 on halitosis assessed by gas chromatography and the levels of proteolytic bacteria determined by quantitative PCR on the tongue coating, in a short term longitudinal study.”

Question: In the central section, should better clarify inclusions and exclusions criteria of the selected patient, with the inclusion/exclusion criteria.

Answer: We agree with the reviewer and have added and highlighted the information in the text, as follows: “Fifty-two individuals of both sexes, recruited at the Dental Clinic of Universidade Nove de Julho, were included. Inclusion criteria: participants from 18 to 25 years old, young adults, with a diagnosis of halitosis exhaling sulfhydride (H2S) ≥ 112 ppb on gas chromatography, seeing as this is the main gas originated from the tongue coating. Exclusion criteria comprised those with dentofacial anomalies (such as cleft lip, palatine, and nasopalatine fissures), undergoing orthodontic and/or orthopedic treatment, undergoing oncological treatment, with systemic alterations (gastrointestinal, renal, hepatic), undergoing antibiotic treatment up to 1 month before the survey, pregnant women, and those with fissured or furrowed tongue. These participants were excluded mainly because these conditions could alter their microbiota, making their data too discrepant to be compared with participants without them.”

Question: The discussion section appears well organized with the relevant paper that support the conclusions, even if the authors should better discuss the relationship regarding the role exerted by NLRP3 and galectin-3 on both halitosis and related periodontitis progression. The conclusion should reinforce in light of the discussions.

Answer: Thank you for your suggestions. We have added the requested information to the Discussion section, as follows: “On the other hand, it is important to remember that periodontitis and its markers are also related to halitosis in the literature. A study showed that patients with periodontitis and periodontitis and diabetes exhibited high serum and salivary NLRP3 inflammasome levels compared to diabetes patients and healthy subjects. Periodontitis and HDL-cholesterol were significant predictors of serum NLRP3 while periodontitis and CRP were the significant predictors of salivary NLRP3. [3] Another article showed that patients with periodontitis and periodontitis + CHD presented significant higher serum and salivary Galectin-3 levels in comparison with CHD patients and healthy subjects. Periodontitis and ET-1 were the significant predictors of serum and salivary Galectin-3 levels, respectively. [4]”

Question: In conclusion, I am sure that the authors are fine clinicians who achieve very nice results with their adopted protocol. However, this study, in my view does not in its current form satisfy a very high scientific requirement for publication in this journal and requests a revision before a futher re-evaluation of the manuscript.

Answer: Thank you much for reviewing our article. All the comments were relevant and have surely helped to improve our paper. 

Question: 

Abstract:

-Better formulate the abstract section by better describing the aim of the study.

Answer: We agree and have completely rewritten and highlighted the Abstract in the manuscript. 

Introduction:

-Please refer to major comments

Discussion

-Please add a specific sentence that clarifies the results obtained in the first part of the discussion

Answer: We have added and highlighted the requested information in the texts, as follows: “The results of the present study showed that there was an immediate reduction in halitosis with aPDT and/or scraper in the clinical evaluation (halimetry). However, probiotics showed no difference in relation to the initial levels. The amount of Porphyromonas gingivalis, Tannerella forsythia and Treponema denticola were not altered throughout the analysis.”

Reviewer #2: Participants diagnosed with halitosis exhaling sulfhydride >= 112 ppb were randomized to one of 4 treatments. The results are unclear.

Answer: Firstly, we would like to thank the reviewer for all the comments and suggestions, which will surely help to improve our article.

Major revisions:

1- Abstract: Write the abstract in complete flowing sentences. Clarify the objective of the study. Briefly name the statistical testing methods applied and state the p-values that support the conclusions.

Answer: We agree and have completely rewritten and highlighted the Abstract in the manuscript. 

2- Line 115: There appears to be an error with the formatting of one of the dates.

Answer: Thank you for warning us. We have corrected the information, following the date format that is used in Clinical Trials, as follows: “It was registered in ClinicalTrials.gov, under the number NCT03996044, first posted on 2019-06-21and last updated on 2023-08-18.”

3- Line 128: State and justify the study’s target sample size with a pre-study statistical power calculation. The power calculation should include: (1) the estimated outcomes in each group; (2) the α (type I) error level; (3) the statistical power (or the β (type II) error level); (4) the target sample sizes and (5) the statistical testing method (6) for continuous outcomes, the standard deviation of the measurements.

Answer: The following paragraph was added to the manuscript: “The sample size was calculated based on the data reported by Costa da Mota et al. [13]. The sample size was estimated using an F test model with 4 groups, a significant level of α = 0.05 and an estimated test power of 1- β 80%.The initial sample size estimation was 11 subjects per group. However, to account for the possible non-parametric distribution of the data, 15% more subjects were added to each group. group [Erich L. Lehmann, Nonparametrics: Statistical Methods Based on Ranks, Revised, 1998, ISBN=978-0139977350, pages 76-81.] resulting in 13 subjects per group. The calculation was caried out using G*Power V. 3.1.97”. 

4- Lines 150-9: Clarify the randomization process. Be more specific and less wordy.

Answer: We have clarified and simplified the randomization explanation, as follows: “Patients were randomized by block randomization into the groups (www.randomizer.at), according to the treatment to be performed. Opaque envelopes were identified with each number and inside it a sheet containing the information of the corresponding experimental group was inserted. The envelopes were opened in order, showing the group to which the participant would belong to when they were included.”

5- Line 297: The data rather than the groups were tested for normality using the Shapiro-Wilk test.

Answer: We apologize for the mistake and have made that correction in the text. 

6- Line 297: The following sentence is unclear. What variables were tested using the Kruskal-Wallis or ANOVA tests? An underlying assumption of the ANOVA is normality of data. “As the data showed no normality, they were treated using the Kruskal-Wallis ANOVA test (independent measures) and Friedman ANOVA (dependent measures) followed by appropriate posthoc tests, when necessary.” State the statistical methods used for the posthoc tests.

Answer: In relation to the initial comments, the Kruskal-Wallis test goes by several names, including the Kruskal–Wallis test by ranks, Kruskal–Wallis H test, one-way ANOVA on ranks, or simply Kruskal-Wallis ANOVA. The inclusion of "ANOVA" in the title emphasizes the comparison of multiple groups, similar to the purpose of ANOVA, but it clarifies that the Kruskal-Wallis test is a non-parametric method. To summarize, using "Kruskal-Wallis ANOVA" signifies that this test is employed to compare multiple groups, akin to ANOVA, but through a non-parametric approach based on ranks. However, we recognize that this terminology might lead to some confusion for the intended readers of this work. As a result, we opted for a simpler term, "Kruskal-Wallis test," to provide clarity. As for the second part of the comments, the specific post-hoc test was added in the manuscript.

7- For data that has been collected repeatedly, use a repeated measures statistical approach for analyzing the results.

Answer: In our analysis of this type of data, we employed the Friedman ANOVA test, as mentioned in the manuscript. The Friedman test proves to be highly useful when working with data that doesn't adhere to a normal distribution and is related, particularly in cases where traditional parametric tests may not be suitable. It enables a strong comparison among multiple groups, aligning well with our experimental designs. Should the reviewer have a specific analysis in mind, we welcome the suggestion and will make every effort to integrate it into the manuscript.

8- Include a comprehensive and specific statistical analysis section, listing and detailing all the statistical methods used.

Answer: All statistical methods have been described and highlighted in the text. 

9- Express p-values more precisely than p > 0.05. 

Answer: We apologize for not citing the values previously. The lacking p values, related to the microbiological analysis have been added and highlighted in the text. 

10- Cite the statistical software used for the analysis.

Ans: For this work we used SPSS V. 25.0 for the inferential analysis and G*Power V. 3.1.97 for the sample size calculation. This information was added to the manuscript. 

11- Thoroughly proofread the manuscript.

Answer: The article was thoroughly revised. Thank you much for reviewing our article. All the comments were relevant and have surely helped to improve our paper.

---

## [Decision Letter · Decision Letter 1]

14 Nov 2023

PONE-D-23-25648R1Short term effect of antimicrobial photodynamic therapy and probiotic L. salivarius WB21 on halitosis: a controlled and randomized clinical trialPLOS ONE

Dear Dr. Kalil Bussadori,

Thank you for submitting your manuscript to PLOS ONE. After careful consideration, we feel that it has merit but does not fully meet PLOS ONE’s publication criteria as it currently stands. Therefore, we invite you to submit a revised version of the manuscript that addresses the points raised during the review process.

We look forward to receiving your revised manuscript.

Kind regards,

Gaetano Isola, Ph.D.

Academic Editor

PLOS ONE

Journal Requirements:

Reviewers' comments:

Reviewer's Responses to Questions

**Comments to the Author**

1. If the authors have adequately addressed your comments raised in a previous round of review and you feel that this manuscript is now acceptable for publication, you may indicate that here to bypass the “Comments to the Author” section, enter your conflict of interest statement in the “Confidential to Editor” section, and submit your "Accept" recommendation.

Reviewer #1: All comments have been addressed

Reviewer #2: (No Response)

2. Is the manuscript technically sound, and do the data support the conclusions?

Reviewer #1: Yes

Reviewer #2: Yes

3. Has the statistical analysis been performed appropriately and rigorously? 

Reviewer #1: Yes

Reviewer #2: No

4. Have the authors made all data underlying the findings in their manuscript fully available?

Reviewer #1: Yes

Reviewer #2: Yes

5. Is the manuscript presented in an intelligible fashion and written in standard English?

Reviewer #1: Yes

Reviewer #2: Yes

6. Review Comments to the Author

Reviewer #1: The authors have well addressed all issues raised by the reviewer. No further adjustments are needed

Reviewer #2: Major Revisions:

1- Abstract: Please revise the following statement to improve clarity. "The clinical data showed no normality, they were treated using the Kruskal-Wallis test ..." Consider the following. "The clinical data failed to follow a normal distribution; therefore, comparisons were made using the Kruskal-Wallis test..."

2- Abstract and Lines 350-352: If the microbiological data was normally distributed, why was a non-parametric test, the Kruskal-Wallis rank sum test used? An analogous parametric test such as an ANOVA would be more powerful if the distribution of the data is normal.

3- Abstract, Results: Provide a p-value to support the primary result.

4- The authors should consider providing tests of interaction effects, specifically time by group interaction effects rather than repeatedly applying Kruskal-Wallis tests.

Minor Revisions:

1- The first paragraph under the "Results" section should be renamed "Statistical analyses".

7. PLOS authors have the option to publish the peer review history of their article (what does this mean?). If published, this will include your full peer review and any attached files.

Reviewer #1: No

Reviewer #2: No

---

## [Author Response · Author response to Decision Letter 1]

3 Dec 2023

Response to Reviewers

Reviewer #1: The authors have well addressed all issues raised by the reviewer. No further adjustments are needed.

Answer: We would like to thank the reviewer once again. 

Reviewer #2: Major Revisions:

1- Abstract: Please revise the following statement to improve clarity. "The clinical data showed no normality, they were treated using the Kruskal-Wallis test ..." Consider the following. "The clinical data failed to follow a normal distribution; therefore, comparisons were made using the Kruskal-Wallis test..."

Answer: Firstly, we would like to thank the reviewer for revising our manuscript once again. The suggested correction was made and highlighted in the text. 

2- Abstract and Lines 350-352: If the microbiological data was normally distributed, why was a non-parametric test, the Kruskal-Wallis rank sum test used? An analogous parametric test such as an ANOVA would be more powerful if the distribution of the data is normal.

Answer: Actually, data distribution was not normal. However, we do agree with the reviewer, the writing was not clearly expressing that. Therefore, we have changed and highlighted the sentence in the text, as follows: “For the microbiological data, seeing as the data failed to follow a normal distribution, the Kruskal-Wallis rank sum test was performed with Dunn's post-test.”

3- Abstract, Results: Provide a p-value to support the primary result.

Answer: p-values have been added and highlighted in the Abstract. 

4- The authors should consider providing tests of interaction effects, specifically time by group interaction effects rather than repeatedly applying Kruskal-Wallis tests.

Answer: Other tests have been used in our paper, such as Friedman ANOVA (for dependent measures) followed by appropriate posthoc tests, when necessary (Dunn’s test for independent measurements and pairwise Wilcoxon signed-rank test with Ryan-Holm stepdown Bonferroni procedure for adjustment). In the microbiological results analysis, the Kruskal-Wallis rank sum test was also followed by the Dunn's post-test. The application of these tests has been mentioned both in the Results and in the Abstract of the manuscript. 

Minor Revisions:

1- The first paragraph under the "Results" section should be renamed "Statistical analyses".

Answer: The correction has been made and highlighted in the text.

---

## [Decision Letter · Decision Letter 2]

15 Dec 2023

PONE-D-23-25648R2Short term effect of antimicrobial photodynamic therapy and probiotic L. salivarius WB21 on halitosis: a controlled and randomized clinical trialPLOS ONE

Dear Dr. Kalil Bussadori,

Thank you for submitting your manuscript to PLOS ONE. After careful consideration, we feel that it has merit but does not fully meet PLOS ONE’s publication criteria as it currently stands. Therefore, we invite you to submit a revised version of the manuscript that addresses the points raised during the review process.

We look forward to receiving your revised manuscript.

Kind regards,

Gaetano Isola, Ph.D.

Academic Editor

PLOS ONE

Additional Editor Comments:

The authors have well addressed to all issues raised by the reviewer. The manuscript can now be acceptable for publication.

Reviewers' comments:

Reviewer's Responses to Questions

**Comments to the Author**

1. If the authors have adequately addressed your comments raised in a previous round of review and you feel that this manuscript is now acceptable for publication, you may indicate that here to bypass the “Comments to the Author” section, enter your conflict of interest statement in the “Confidential to Editor” section, and submit your "Accept" recommendation.

Reviewer #1: All comments have been addressed

Reviewer #2: (No Response)

2. Is the manuscript technically sound, and do the data support the conclusions?

Reviewer #1: Yes

Reviewer #2: No

3. Has the statistical analysis been performed appropriately and rigorously? 

Reviewer #1: Yes

Reviewer #2: No

4. Have the authors made all data underlying the findings in their manuscript fully available?

Reviewer #1: Yes

Reviewer #2: Yes

5. Is the manuscript presented in an intelligible fashion and written in standard English?

Reviewer #1: Yes

Reviewer #2: Yes

6. Review Comments to the Author

Reviewer #1: In the present study, the authors have well addressed all issues raised by the reviewer. The manuscript can be accepted.

Reviewer #2: Major revisions:

Line 325: Instead of performing between group comparisons at each time point individually. Use a model that allows for testing the interaction of time by treatment effect. A comprehensive reanalysis is needed.

Process for testing interaction effects:

If the interaction effect is significant, provide an interpretation of the results, but do not test main effects because the tests for main effects are uninteresting in light of significant interactions. If interaction effects are non-significant, drop the interaction effects from the model and test the main effects. Determining which results to present when testing interactions is often a multi-step process.

Minor revision:

1- Line 316: Consider replacing "treated" with "compared".

Note line numbers refer to those in the tracked changes version of revision 1.

7. PLOS authors have the option to publish the peer review history of their article (what does this mean?). If published, this will include your full peer review and any attached files.

Reviewer #1: No

Reviewer #2: No

---

## [Author Response · Author response to Decision Letter 2]

20 Dec 2023

Response to Reviewers

Reviewer #1: In the present study, the authors have well addressed all issues raised by the reviewer. The manuscript can be accepted.

Answer: We would like to thank the reviewer once again. 

Reviewer #2: Major revisions:

Line 325: Instead of performing between group comparisons at each time point individually. Use a model that allows for testing the interaction of time by treatment effect. A comprehensive reanalysis is needed.

Process for testing interaction effects:

If the interaction effect is significant, provide an interpretation of the results, but do not test main effects because the tests for main effects are uninteresting in light of significant interactions. If interaction effects are non-significant, drop the interaction effects from the model and test the main effects. Determining which results to present when testing interactions is often a multi-step process.

Answer: We sincerely appreciate your insightful feedback on the statistical analysis approach employed in our study. Your comments have provided valuable insights into alternative methodologies, and we carefully consider your recommendations.

It is acknowledged that the Two-way repeated measures ANOVA test is well-suited for research designs similar to ours but that present an approximately parametrically distributed data. However, given the non-parametric nature of our dataset, applying this method directly may not be appropriate, as it assumes different characteristics inherent in non-parametric data. Other alternative approaches, such as Mixed Effects Models or Multilevel Models also rely on the assumption of normality of residuals. Unfortunately, this assumption cannot be guaranteed by our data.

In light of the non-parametric nature of our data, we opted for the use of the Friedman Test and Kruskal-Wallis Test. These methods are specifically designed to accommodate the non-normal distribution and ranking structure inherent in non-parametric data. To maintain the overall significance level at 0.05, we recognize the importance of implementing corrections for local p-values, such as the Bonferroni or the Ryan-Holm stepdown Bonferroni procedure, even though they may impact test power. Additionally, any interaction within or between groups would have been detected by such tests and would also be observed in the descriptive statistics.

Considering the specific outcomes of our analysis, we observed high p-values (> 0.60) in the between-groups analysis, consistent with our clinical observations. Even with a different statistical approach, these p-values would likely remain above 0.05, thereby preserving the study's conclusions. Similarly, the "Before" and "After" analysis yielded very low p-values (p < 0.001), and alternative statistical approaches are unlikely to alter these results significantly.

We also acknowledge that the chosen statistical models are well-recognized and understood by our target audience. Introducing a more uncommon model might add complexity without commensurate benefits in terms of the level of certainty and overall information provided in the work.

Should you have specific suggestions for a statistical model that could enhance the readability and comprehension of our work while aligning with the assumptions of our data, we would be more than willing to discuss and explore those possibilities.

Minor revision:

1- Line 316: Consider replacing "treated" with "compared".

Answer: The change has been made and highlighted in the text, as requested.

We would like to thank the reviewer, for revising our paper once again.

---

## [Decision Letter · Decision Letter 3]

3 Jan 2024

Short term effect of antimicrobial photodynamic therapy and probiotic L. salivarius WB21 on halitosis: a controlled and randomized clinical trial

PONE-D-23-25648R3

Dear Dr. Kalil Bussadori,

We’re pleased to inform you that your manuscript has been judged scientifically suitable for publication and will be formally accepted for publication once it meets all outstanding technical requirements.

Kind regards,

Gaetano Isola, Ph.D.

Academic Editor

PLOS ONE

Additional Editor Comments (optional):

The manuscript is acceptable in this current form.

Reviewers' comments:

Reviewer's Responses to Questions

**Comments to the Author**

1. If the authors have adequately addressed your comments raised in a previous round of review and you feel that this manuscript is now acceptable for publication, you may indicate that here to bypass the “Comments to the Author” section, enter your conflict of interest statement in the “Confidential to Editor” section, and submit your "Accept" recommendation.

Reviewer #1: All comments have been addressed

Reviewer #2: All comments have been addressed

2. Is the manuscript technically sound, and do the data support the conclusions?

Reviewer #1: Yes

Reviewer #2: (No Response)

3. Has the statistical analysis been performed appropriately and rigorously? 

Reviewer #1: No

Reviewer #2: (No Response)

4. Have the authors made all data underlying the findings in their manuscript fully available?

Reviewer #1: Yes

Reviewer #2: (No Response)

5. Is the manuscript presented in an intelligible fashion and written in standard English?

Reviewer #1: Yes

Reviewer #2: (No Response)

6. Review Comments to the Author

Reviewer #1: In this revised version of the manuscript, the authors have made all requested changes. No further issues are needed.

Reviewer #2: (No Response)

7. PLOS authors have the option to publish the peer review history of their article (what does this mean?). If published, this will include your full peer review and any attached files.

---

## [Editor Report · Acceptance letter]

21 Jun 2024

PONE-D-23-25648R3 

PLOS ONE

Dear Dr. Kalil Bussadori, 

I'm pleased to inform you that your manuscript has been deemed suitable for publication in PLOS ONE. Congratulations! Your manuscript is now being handed over to our production team.

Kind regards, 

on behalf of

Prof. Gaetano Isola 

Academic Editor

PLOS ONE